# Is the COVID-19 Pandemic Over? The Current Status of Boosters, Immunosenescence, Long Haul COVID, and Systemic Complications

**Miriam Ting** [1,*] and **Jon B. Suzuki** [2,3,4,5]

1    Think Dental Learning Institute, Paoli, PA 19301, USA
2    University of Maryland, Baltimore, MD 20742, USA; jon.suzuki@temple.edu
3    University of Washington, Seattle, WA 98195, USA
4    Nova Southeastern University, Fort Lauderdale, FL 33314, USA
5    Temple University, Philadelphia, PA 19140, USA
*    Correspondence: thinkdentallearninginstitute@gmail.com; Tel.: +1-610-601-8898

**Abstract:** The coronavirus disease 2019 (COVID-19) pandemic, caused by severe acute respiratory syndrome coronavirus 2 (SARS-CoV-2), appears to be diminishing in infectivity and hospitalizations in the United States and many parts of the world. This review will provide current information on the pathogenesis of SARS-CoV-2 and long haul COVID, emerging research on systemic complications, and antibody responses of vaccines and boosters.

**Keywords:** COVID-19; SARS-CoV-2; coronavirus; immunosenescence; vaccines; boosters

## 1. Introduction

Coronavirus disease 2019 (COVID-19) was first reported on 31 December 2019, and by 11 March 2020, it was declared a global pandemic by the WHO. COVID-19 started in Wuhan (China) and is caused by the highly contagious severe acute respiratory syndrome coronavirus 2 (SARS-CoV-2). This single stranded RNA virus has cell-surface spike glycoproteins which penetrate and adhere to host cells [1]. Entry into the host cell is via the angiotensin-converting enzyme 2 (ACE-2) receptor, which is found in the heart, lungs, kidneys, tongue, and salivary glands [1]. SARS-CoV-2 can easily colonize oral, nasal, and pharyngeal mucosa [2]. Transmission of SARS-CoV-2 occurs via aerosol, droplet, oral–fecal routes [3], and contaminated body fluids and surfaces [4].

Clinical COVID-19 symptoms included fever, dry cough, sore throat, myalgia, fatigue, diarrhea [5,6], and loss of taste [7]. These symptoms may appear 5.2 days after infection [8]. The majority of the time, COVID-19 infected patients may be asymptomatic or have mild symptoms. The report of acute respiratory distress syndrome (ARDS) or multi-organ failure was less than 5% [8]. SARS-CoV-2 can be highly contagious; asymptomatic patient may also transmit the virus. A study reported that COVID-19 transmission in asymptomatic patients and symptomatic patients were statistically similar [9]. The risk factors for COVID-19 include advanced age, diabetes, hypertension, obesity, and heart disease [10–12].

Healthcare facilities, including medical and dental offices, are at risk for cross infection between healthcare professionals and patients [13]. This risk of spread can be mitigated by the use of personal protective equipment (PPE) including masks, face shields, and gowns, and by preventive strategies including hand washing and pre-procedural mouth rinsing [14].

## 2. Pathogenesis and Immunosenescence

COVID-19 progression includes: (1) innate immunity activation; (2) adaptive immunity activation; (3) cytokine release syndrome ("cytokine storm") [15]. Cytokine storms (Figure 1) are the result of a hyper-responsive host producing exaggerated cytokine release [16,17]. Cytokine storms increase vascular permeability and effector cell infiltration, resulting in excessive monocyte proliferation, lymphocyte apoptosis, and immunodeficiency states [18]. The clinical outcomes include shock, multiple organ dysfunction, hypercoagulation, acute lung injury [15], and multi-organ failure, including the kidneys and heart [19–21].

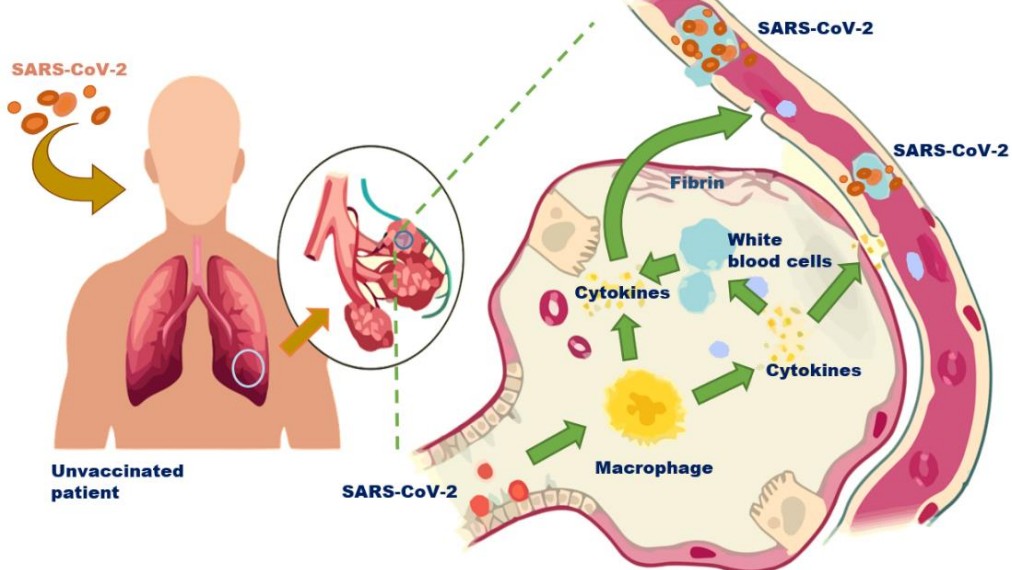

**Figure 1.** SARS-CoV-2 pathogenesis in an unvaccinated patient illustrating cytokine storm impact of damage to lungs and adjacent blood vessels.

The COVID-19 proinflammatory factors involved in the cytokine storm included IFN-γ, IFN-γ-induced protein 10, IL-1, IL-6, IL-12, and monocyte chemoattractant protein [22]. COVID-19 non-survivors present with higher IL-6 levels than survivors [17]. IL-6 has been linked to increased severity [23–27] and deaths in the elderly and immunocompromised [28]. This increased inflammatory cytokine activation can cause long-lasting damage to the immune system [29]. The formation of microthrombi to larger blood clots in the vessels of major organs, including the lungs, may be responsible for the debilitating systemic effects of COVID-19 on the body [29].

Oral health and systemic health may influence COVID-19 susceptibility. The antibody response to SARS-CoV-2 peaks at 14–21 days after COVID-19 exposure [30]. SARS-CoV-2 can also stimulate neutralizing secretory antibodies, Immunoglobulin A (IgA), which can dominate the initial mucosal immune response in the oral cavity [31]. Patients with periodontal inflammation or other chronic inflammatory diseases, with an incipient heightened proinflammatory response, may have an increased risk of SARS-CoV-2 susceptibility and complications. In the oral cavity, the periodontal response to bacterial were designated as "high", "low", and "slow" [32]. High IL-1β levels were detected in the inflamed tissues of the "high" group. These differences of inflammatory response may contribute to COVID-19 patients, having different levels of disease severity from mild infections, hospitalizations, or morbidity [32].

COVID-19 infected older adults aged 70 and above presented with multiple complications and mortality [33]. Severe complications of COVID-19 were septic shock, blood clots, sepsis, pneumonia, and ARDS [34]. The cause of death was not usually the initial viral infection, but post-viral complications like ARDS. Headaches, encephalitis, and strokes

have been reported complications in patients with COVID-19 [35]. Cardiac signs and symptoms include heart damage, arrhythmias, and heart failure. Myocarditis and cardiac muscle inflammation have been reported complications of COVID-19 [35].

Age-related compromised immunity (immunosenescence) may be the cause of increased mortality from COVID-19 in the elderly. Immunosenescence affects innate and adaptive immunity; it may cause increased cytokine production [36], lymphocyte blastogenesis impairment [37], ineffective antibody production, failed T-cell response, and severe inflammatory organ dysfunction [38]. Thus, immunosenescence may increase the susceptibility and the severity of COVID-19, as well as diminish the responses to the vaccine. This may result in higher COVID-19 vaccine breakthrough infections [39]. In August 2021, the Centers for Disease Control and Prevention (CDC) reported COVID-19 breakthrough infections, which could be due to waning vaccine antibody reaction or emerging SARS-CoV-2 variants [40].

## 3. Long Haul COVID

The healing time for COVID-19 is approximately 2 weeks for mild disease, and 3–6 weeks for more severe infections [41]. COVID-19 may progress to long haul COVID when symptoms extend beyond 4 weeks. Approximately 25–40% of COVID-19 infected patients will progress to long haul COVID [42]. The prevalence of long haul COVID were reported as follow: USA 16–53% [43,44], UK 1.6–71% [45–48], Denmark 1% [49], Germany 35–77% [50,51], Italy 5–51% [52,53], China 49–76% [54,55], Africa 68% [56], Bangladesh 16–46% [57,58], and India 22% [59,60]. A retrospective study reported that 34% of COVID-19 patients had lingering psychiatric or neurological symptoms 6 months after COVID-19 [42]. Another study reported that 87.4% of hospitalized COVID-19 patients have persistent symptoms after 60 days [61]. A Danish survey of 152,000 people reported that almost one third of the people surveyed had at least one persistent symptom between 6 and 12 months after COVID-19 onset. This survey revealed that the most commonly reported long-term symptoms were fatigue and impairment of taste and smell. These symptoms related to long haul COVID can last for at least 12 weeks [62]. Hospitalized patients reported higher prevalence compared to community patients [63].

Long haul COVID differs from acute COVID-19. Long haul COVID patients are survivors of acute COVID who have developed persistent symptoms that last for at least 6 months [64]. Long haul COVID can affect COVID survivors of all severity and age. Long haul COVID can also affect survivors who are no longer SARS-CoV-2 positive [65]. Following acute COVID-19 infection, a possible mechanism for long haul COVID could be the chronic inflammatory responses to persistent viral reservoirs [66] or the delayed damage from the autoimmune response to host antigens via molecular mimicry [67].

Long haul COVID has a higher prevalence in women and in patients aged 24–36 years [68]. Risk factors reported for long haul COVID include age, smoking, asthma, obesity, poor health, autoimmune diseases, and chronic inflammatory diseases [14,69]. Pre-existing asthma is significantly associated with long haul COVID [45]. Obese patients are 25% more likely to progress to long haul COVID than patients that are not obese [70]. Long haul COVID symptoms may include fever, fatigue, brain fog, headaches, dyspnea, coughing, nausea, vomiting, anxiety, depression, muscle pain, chest pain, skin rash, palpitations, post-exertional malaise, and joint pain [71]. In a social media survey of COVID patients, 89% reported persistent cardiopulmonary symptoms [72]. Other symptoms may include anxiety, depression, psychosis, venous thromboembolism, as well as cardiac, hepatic, and renal impairment [73]. A UK study [48] of hospitalized patients at 5 months post-discharge reported 48% with persistent fatigue, 41% dyspnea, and 21–28% with chest pain and palpitations. A China study [54] of hospitalized patients at 6 months post-infection reported 63% with fatigue, 26% dyspnea, and 5–9% with chest pain and palpitations. This China study [55] at 12 months further reported improvements in this patient group to 30% with dyspnea, 7% with chest pain, and 20% with fatigue. Taste and smell impairment [20], gastrointestinal disturbances like nausea, and loss of appetite, diarrhea, and bowel blockages

that were reported in acute COVID-19 were not consistently reported in long COVID. Systemic conditions arising from acute COVID-19 may persist during long haul COVID [74]. The increased inflammatory cytokines initiated by SARS-CoV-2 can result in prolonged immune system damage [29].

Coughing, dyspnea, and fatigue were consistently reported in long haul COVID. These symptoms may be related to the persistent cytokine production by pulmonary inflammatory cells. The elevated cytokines in long haul COVID include IL 1-β, TNF-α, IL-6, among others [29]. A prospective study evaluated the serum analytes from patients with long haul COVID for over 8 months and reported that these patients had highly activated innate immunity lacking in naïve T and B cells, and increased expression of type I and Type III interferon [62]. Persisting immune activation may be due to lingering antigens, autoimmunity, or impaired healing [62]. Patients with long haul COVID-19 also reported palpitations and angina [35]. This increased risk of headaches, encephalitis, and strokes in patients with long haul COVID may require constant monitoring [35].

Although children may have less severe COVID-19 than adults [75], long haul COVID and multisystem inflammatory syndrome has been reported as a long-term consequence of SARS-CoV-2 infection in children. Both long-term consequences can even affect asymptomatic COVID-19 infected children [76]. The prevalence of long haul COVID reported in a systematic review was 25.24% and the most common clinical manifestations were mood symptoms, fatigue, and sleep disorders [77].

The patient's levels of D-dimer, C-reactive protein (CRP), and lymphocytes were potential inflammatory biomarkers of long haul COVID. Increased levels of D-dimer, CRP, and reduced lymphocytes were more common in patients with persistent symptoms than fully recovered patients [78]. These systemic inflammatory biomarkers were also associated with radiological abnormalities of the heart, liver, and kidney at 2–3 months following discharge of COVID-19 patients [79].

Healthcare professionals should be aware that long COVID is becoming increasingly more prevalent. Oral healthcare professionals should be prepared to treat these patients safely in an outpatient setting per CDC infection control guidelines [73]. Potential treatment for long haul COVID may include rehabilitation, behavioral modification, psychological support, or pharmacologic treatments. Rehabilitation may include light aerobic exercise that gradually increases in intensity until improvements are seen [80]. Behavioral modification and psychological support aim to improve wellness and mental health [80]. In elderly post-COVID patients, a randomized controlled trial reported that a 6 week rehabilitation program improved exercise tolerance, lung functions, quality of life, and anxiety [81]. However, rehabilitation may not be suitable for post-COVID patients with severe lung or cardiac damage, and in situations where exercise is contraindicated [82]. Presently, no pharmacologic medicine has been shown to have significant effects on long haul COVID. However, anti-inflammatory drugs may be used to manage long haul COVID-specific symptoms like fever and pain [83]. Shared pathophysiology of long haul COVID and postural orthostatic tachycardia syndrome (POTS) suggests potential drug repurposing. A study of 24 post-COVID patients with palpitations reported that a POTS medication (Ivabradine) effectively relieved palpitations [84]. Other pharmacologic therapies that are further investigated for repurposing are as follows: metabolic modulators (Niagen), immunomodulatory therapies (Steroids, laranilubmab, Tocilizumab, Atorvastatin, Colchicine), antifibrotic treatments (Pirfenidone, LYT-100), and anticoagulation (Apixaban) [63]. The use of these medications is preliminary and more research is needed to confirm efficacy.

Vaccines may provide significant protection against COVID-19 breakthrough infections and long haul COVID [85]. In a United Kingdom study [86], there was a reduced risk for long-haul COVID, serious complications, and breakthrough infections in fully vaccinated patients.

## 4. Vaccines and Boosters

The US Food and Drug Administration (FDA) guidance document stated that for a vaccine to be considered it should have at least 50% efficacy [87]. Vaccine effectiveness is proportional to the reduction of infection between the vaccinated and non-vaccinated subjects. The ideal vaccine would need to be effective after 1–2 doses, with at least 6 months of protection, and reduce transmission in the infected. It should prevent infection and disease transmission, as well as reduce mortality and disease severity. Randomized controlled vaccine trials evaluated reduction of clinical disease severity, infection, and infectivity duration [88]. However, socioeconomic conditions, geographical settings, age differences, and herd immunity may interfere with the data.

Vaccine development encompasses many methodologies, including targeted nucleic acid DNA or mRNA, adenovirus carrier (viral vector), spike proteins (protein subunits), and inactivated (whole) virus [89]. The objective of these vaccines is to neutralize the mRNA, spike protein, or the virus [90]. A robust Ig G response by B-lymphocytes and plasma cells initiated by these vaccines provides adequate immunological defense against invading SARS-CoV-2 (Figure 2).

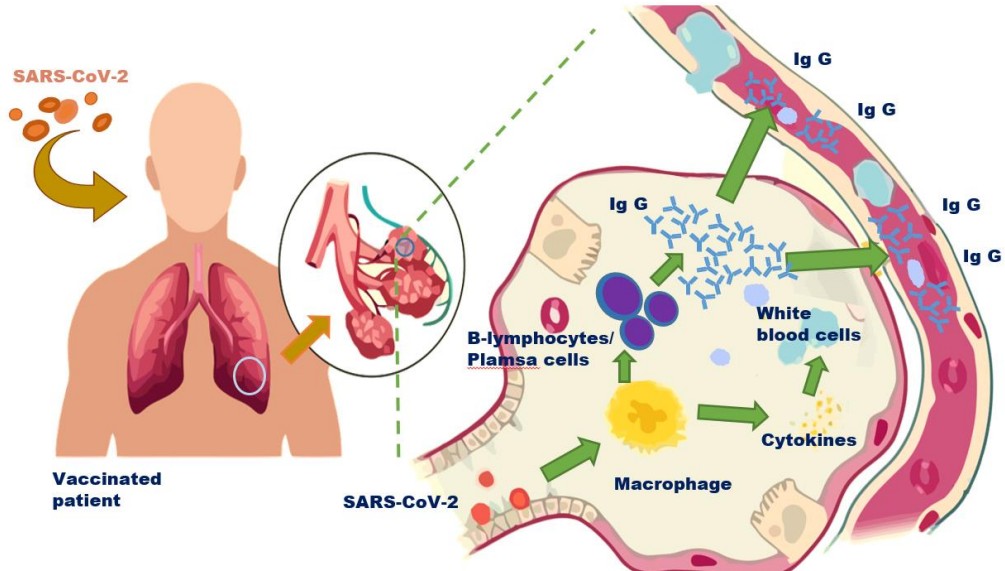

**Figure 2.** Immune response and IgG antibodies against SARS-CoV-2 reduced in waning fully vaccinated patient or patient with immunosenescence.

Clinical trials for SARS-CoV-2 vaccines evaluated the antibody response for the following: adenoviral vector [91,92], mRNA [93,94], spike glycoprotein [95], and inactivated SARS-CoV-2 [96,97]. Some of the major vaccines are outlined in Table 1.

This UK report supports patients receiving both doses of the two-dose vaccine regimen, with 94% of patients remaining asymptomatic after both doses. Fourteen days after the first dose (Pfizer-BioNTech, Moderna, or AstraZeneca–Oxford vaccines), patients have a 0.5% risk of a breakthrough infection. This dropped to 0.2% of patients with COVID breakthrough infection after the second dose of these vaccines [86].

The diminishing immunologic memory of the patient (Figure 3) or the mutating antigenicity of SARS-CoV-2 may decrease in vaccine efficacy as time progresses. A study showed 64% vaccine effectiveness in long term care residents with a median age of 84, and 90% effectiveness in healthcare workers [98]. The vaccine effectiveness in older individuals that are on long term care are more muted compared to healthy older individuals [99]. Vaccine boosters may extend protection, and boosters comprising of multiple vaccinations or with multiple vaccine types may induce a more robust and persistent immunity [100,101]. The medically-compromised have lowered vaccine effectiveness and higher risks of COVID-19

breakthrough infections [39]. However, breakthrough infections have also been reported in fully vaccinated patients [40]. Despite that, vaccines can significantly reduce breakthrough infections. The United Kingdom data [86] reported reduced complications risk, breakthrough infections, and long haul COVID in fully vaccinated patients. However, the antibody levels in the vaccinated may decline faster than the those who have been infected with SARS-CoV-2. A study of 25,000 healthcare workers in United Kingdom reported that infection with SARS-CoV-2 reduced the risk of catching the virus again by 84% for 7 months [102]. For the uninfected but vaccinated individuals, the requirement for a booster would depend on the rate of antibody decline (immunosenescence).

**Table 1.** Major SAR-CoV-2 vaccines available (updated April 2022).

| Type | Vaccine | Age Group | Doses | Booster |
|---|---|---|---|---|
| mRNA | Pfizer-BioNTech | Adults (>18 years) Teens (12–18 years) Children (5–11 years) | 2 (30 ug/mL, 3 weeks apart) (>12 years) (10 ug/mL, 3 weeks apart) (5–11 years) | Yes |
| | Moderna | Adults (>18 years) Teens (12–18 years) Children (6–11 years) Children (0.5–5 years) | 2 (100 ug/mL, 4 weeks apart) (>18 years) TBD (6–18 years) Pending approval | Yes |
| Viral vector | Janssen J & J | Adults (>18 years) | 1 (0.5 mL) | Yes |
| | AstraZeneca-Univ. of Oxford | Adults (>18 years) | 2 (0.5 mL, 8–12 weeks apart) | TBD |

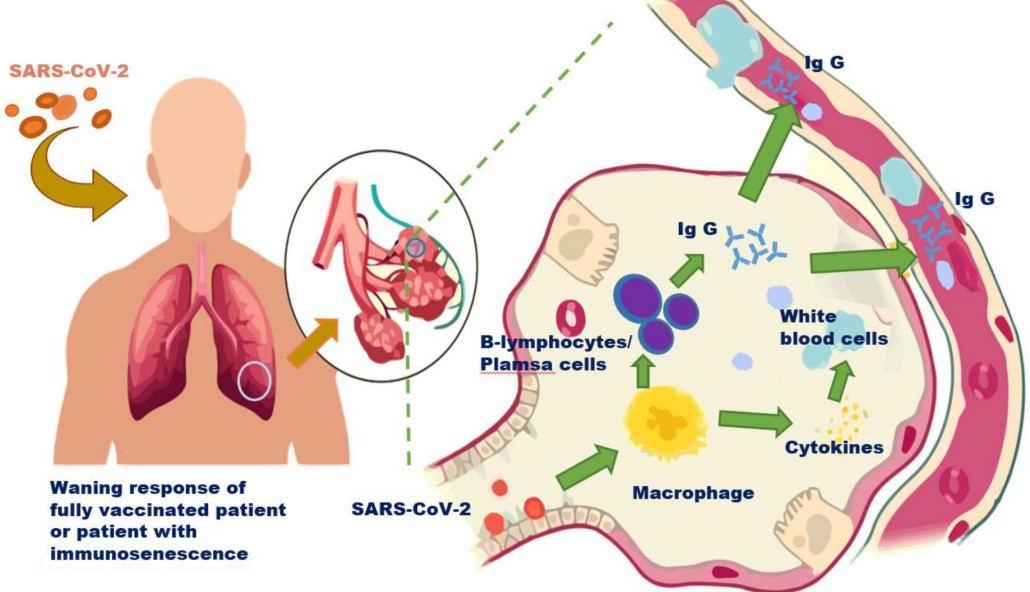

**Figure 3.** Immune response and IgG antibodies against SARS-CoV-2 reduced in waning fully vaccinated patient or patient with immunosenescence.

The European Medicines Agency data in December 2021 suggests boosters following full vaccination of patients. However, currently there is no consensus among clinicians on recommendations for timing of a second booster [103]. Immunocompromised patients, aging patients, and patients with certain systemic diseases and conditions, BMI over 30, and immunosenescence play a role in longevity of antibody protection against SARS-CoV-2 [14].

Immunized individuals also appear to have high levels of neutralizing secretory IgA antibodies against SARS-CoV-2 [104].

Despite the success and safety of the COVID-19 vaccines, very rare but life-threatening cases of thrombosis were reported after ChAdOx1nCov-19 (AstraZeneca) vaccination [105]. This presented as unusual blood clots in unusual anatomical locations, mostly reported as sinus or cerebral thrombosis with thrombocytopenia, and is named Vaccine-associated Immune Thrombosis and Thrombocytopenia (VITT). Of vaccinated cases, VITT was reported between 1 in 125,000 and 1 in 1 million [106]. Onset of symptoms reported approximated 1–2 weeks after vaccination [107]. Treatment for it was mostly unfractionated heparin or sometimes immunomodulatory agents like immunoglobulin or steroids. Mortality rate from VITT was reported as 41.0% [108]. Based on the limited cases reported, females on contraceptives seem to be at the highest risk. However, this is ever-changing as more surveillance safety data becomes available for this and other COVID-19 vaccines. The benefits of the COVID-19 vaccinations outweigh the negative effects and incidence of adverse reactions [109].

## 5. Conclusions

The COVID-19 pandemic appears to be slowly diminishing with the passage of time with enhancement of preventive and therapeutic strategies, like social distancing, good hand washing, and use of antimicrobial mouth rinses. However, evolving clinical research and observations have resulted in additional recognized systemic manifestations, including but not necessarily limited to multiple organ dysfunction, hypercoagulation, acute lung injury, and multi-organ failure, including the kidneys and heart. These systemic complications associated with COVID-19 may have lingering effects with long haul COVID patients. Immunosenescense may limit the antibody response against SARS-CoV-2 and contribute to "breakthrough infections" despite vaccinations. Vaccines and boosters against SARS-CoV-2 and optimal systemic and oral health may prevent the spread of COVID-19 and increase survival. Current data for appropriate booster intervals is contingent on existing, recognized risk factors of vaccinated patients coupled with rate and extent of immunosenescense.

**Author Contributions:** M.T. participated in the design of the review, article selection and literature search, drafting revision of manuscript and figures, and final approval of submitted version. J.B.S. participated in the design of the review, article selection, drafting and revision of manuscript and figures, and final approval of submitted version. All authors have read and agreed to the published version of the manuscript.

**Funding:** This research received no external funding.

**Institutional Review Board Statement:** Not applicable.

**Informed Consent Statement:** Not applicable.

**Data Availability Statement:** Data is contained within the article.

**Conflicts of Interest:** The authors declare no conflict of interest in the writing of this manuscript. J.B.S. is a U.S. Government Special Government Employee with the US Food and Drug Administration, Silver Spring, MD, USA.

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
