# Peer review of "Is the COVID-19 Pandemic Over? The Current Status of Boosters, Immunosenescence, Long Haul COVID, and Systemic Complications"

_2673-8937, doi:10.3390/ijtm2020021_

Round 1
Reviewer 1 Report
This paper proposes a broad and accurate analysis of the pathogenetic mechanisms of Covid 19 infection, taking up the aspects underlying the development of the disease and in particular of organ damage mediated by cytokines. These mechanisms are analyzed in the unvaccinated patient and in the patient with reduced post-vaccination antibody coverage or limited by involution of the immune system.
The topics are well developed, the bibliographic references are cited in the text with appropriate emphasis and the bibliographic entries are updated.
The issue of vaccines is then dealt with; in this regard, in our opinion, it would be opportune to also fully address the issue of potential thrombotic complications, described in particular with adenovirus-type vector vaccines. Another issue that, from a clinical point of view, must in our opinion be mentioned in the paragraph relating to vaccines is represented by the possible finding of acute myocarditis forms in the post-vaccination phase.
I
Author Response
Thank you for the insightful comments. The potential thrombotic complications and other cardiac related complications were added to the manuscript
Reviewer 2 Report
The review by Ting and Suzuki discusses several aspects of
Immunosenescence, long-haul COVID-19 and systemic complications. Although the authors have touched upon topics such as Immunosenescence, long-haul COVID-19 and current vaccines, each of these topics needs to be covered in greater depth and detail, which is currently missing. The authors should have included patient studies and treatment approaches to immunosenescence and long-haul COVID-19 in greater detail. Since there are several reviews already published on the long-term effects of COVID-19, the authors need to provide novel insights or information in their review that has not been presented elsewhere.
The following points need to be covered in greater detail-
- How is the pathogenesis and cellular mechanism of long-haul COVID different from acute COVID-19?
- How is the treatment approach different for long-haul COVID different from acute COVID-19? It would be nice if the authors can review the causes of long COVID syndrome and suggest ways that can provide a basis for a better understanding of the clinical symptomatology for improved diagnostic and therapeutic procedures for the condition.
- Studies on risk factors related to long-haul COVID-19 can be included.
Author Response
Thank you for the insightful comments. The 3 points mentioned were covered in greater details and any other newer information we could find were added
Round 2
Reviewer 1 Report
The re-evaluation of the paper with an effective revision and extension of the issues that had been reported , appears to be more adequate in terms of contents and of wider interest in the clinical field. The general layout and the subdivision of the chapters are well characterized. The articles cited and the related bibliographic entries on the subject of the Long haul Covid are complete and are opportunely linked to the first part of pathogenesis. The additions to the next part relating to vaccines are also relevant and complete.
So I believe that the quality of this revision is good and it’s allow a better scientific contribution of this paper
Reviewer 2 Report
The authors have satisfactorily answered the reviewer's comments and the manuscript can be accepted in its current form.